# *Neurospora* sp. Mediated Synthesis of Naringenin for the Production of Bioactive Nanomaterials

**DOI:** 10.3390/bioengineering11050510

**Published:** 2024-05-18

**Authors:** Jitendra Dattatray Salunkhe, Indra Neel Pulidindi, Vikas Sambhaji Patil, Satish Vitthal Patil

**Affiliations:** 1School of Life Sciences, Kavayitri Bahinabai Chaudhari North Maharashtra University, Jalgaon 425001, India; jitds2016@gmail.com; 2Jesus’ Scientific Consultancy for Industrial and Academic Research (JSCIAR), Tharamani 600113, India; 3University Institute of Chemical Technology (UICT), Kavayitri Bahinabai Chaudhari North Maharashtra University, Jalgaon 425001, India; vikasudct@gmail.com

**Keywords:** *Neurospora* sp., biotransformation, naringinase, naringin, hydrolysis, naringenin, Ag, Au, nanoparticles, nanoconjugates, nematicidal

## Abstract

The application of *Neurospora* sp., a fungus that commonly thrives on complex agricultural and plant wastes, has proven successful in utilizing citrus peel waste as a source of naringin. A UV-Vis spectrophotometric method proved the biotransformation of naringin, with an absorption maximum (λ_max_) observed at 310 nm for the biotransformed product, naringenin (NAR). Further verification of the conversion of naringin was provided through thin layer chromatography (TLC). The *Neurospora crassa* mediated biotransformation of naringin to NAR was utilized for the rapid (within 5 min) synthesis of silver (Ag) and gold (Au) nanoconjugates using sunlight to accelerate the reaction. The synthesized NAR-nano Ag and NAR-nano Au conjugates exhibited monodispersed spherical and spherical as well as polygonal shaped particles, respectively. Both of the nanoconjugates showed average particle sizes of less than 90 nm from TEM analysis. The NAR-Ag and NAR-Au nanoconjugates displayed potential enhancement of the antimicrobial activities, including antibacterial and nematicidal properties over either standalone NAR or Ag or Au NPs. This study reveals the potential of naringinase-producing *Neurospora* sp. for transforming naringin into NAR. Additionally, the resulting NAR-Ag and NAR-Au nanoconjugates showed promise as sustainable antibiotics and biochemical nematicides.

## 1. Introduction

Naringenin (NAR), a valuable heterocyclic flavonoid of medicinal value, is naturally produced by plants and fungi, featuring two interconnected benzene rings with a heterocyclic pyrrole ring. A wide range of pharmacological benefits, namely, antiviral [1], antioxidant [2], antibacterial [3], and anticancer properties [4], were well documented. Furthermore, naringenin effectively inhibits acyl-CoA-cholesterol-o-acyltransferase activity, aiding the prevention of hepatic disease caused by the buildup of macrophage–lipid complexes [5]. In addition to its pharmacological benefits, naringenin has been extensively reviewed for its numerous industrial applications [6]. Large-scale production of naringenin can be achieved through two primary methods: chemical hydrolysis and enzymatic processes. Both approaches involve deriving naringenin from naringin, which is abundant in citrus, orange, and grapefruit wastes. Among these methods, enzymatic processes utilize *naringinase* (EC 3.2.140) to catalyze the conversion of naringin.

Comprehensive research has been carried out on both plants and microorganisms that form potential sources of naringinase. While most *Aspergillus* species are known to produce naringinase enzymes, there is still ongoing interest in exploring diverse sources of naringinase that are suitable to meet the safety profiles and regulations of the food industry. Recent studies have demonstrated NAR biosynthesis by *Streptomyces clavuligerus*, which was cultivated in an aromatic amino acid-rich medium (Tryptic Soy Broth, TSB), and *Saccharomyces cerevisiae*, utilizing D-xylose as the sole carbon source [7,8]. These findings highlight the potential of diverse microorganisms for NAR biosynthesis.

*Neurospora* sp. is a genus of *Ascomycete* fungi well-known as a model organism for genetics research. It has garnered significant interest for enzyme production due to its remarkable characteristics, including a high growth rate, with cell mass doubling every 2 to 3 h under controlled conditions. *Neurospora* sp. also exhibits the ability to utilize complex agricultural and forestry residues and can grow on a simple minimal medium, making it easy to control various parameters for enzyme production. 

Additionally, *Neurospora* sp. is used as a food in Indonesia and has been declared safe as a food by the Food and Drug Administration (FDA), US, for its edible characteristics. *Neurospora* sp. is reported to be useful for pectinase enzyme production by utilizing fruit juice processing waste, i.e., citrus and mango. *Neurospora* sp. was employed for the production of naringinase, which is a citrus fruit juice debittering enzyme. The edibility of *Neurospora* sp. is an important characteristic for the food industry and is exempted from all stringent FDA regulations [9]. Further research and investigation into harnessing its enzymatic potential could hold the key to unlocking valuable applications in various industries.

The versatile, characteristic and diverse pharmacological activities of naringenin make it a fascinating subject of research for potential applications in various fields. However, despite its excellent bioactivities, naringenin poses challenges in successfully translating its potential into clinical applications. There are some major hurdles for the wide-ranging exploitation of naringenin, including its poor bioavailability, insolubility, instability towards oxidation, pH intolerance, and high intestinal metabolism [10,11,12]. Addressing these challenges and improving the bioavailability and stability of naringenin is crucial to unlocking its full therapeutic potential. Researchers are actively exploring various strategies to enhance naringenin’s delivery and stability, such as nanoencapsulation or nanoconjugates, formulation with other compounds to improve solubility, and chemical modifications [10,11,12,13]. In this context, recently, our group highlighted the formation of naringenin-silver nanoconjugates which showed enhanced antimicrobial activities against the acanthamoeba and bacterial pathogens [14]. It is well proven that the modification of a natural compound and that functionalized with nano metals helps to enhance its bioactivities [14]. Previously, some reports highlighted the metal nanoconjugates of natural compounds such as nicin-Ag nanoconjugate and peptide-Ag conjugates that exhibited superior antibacterial activity over the activity of standalone compounds [15,16]. In addition to their antibacterial activity, metal nanoconjugates have shown enhanced activity against acanthamoeba, demonstrating their remarkable potential for combating various microbial pathogens [14]. This promising finding suggests that further research and exploration of nanoconjugates should be conducted to assess their efficacy against a wider range of microbial pathogens. Nematodes are economically significant phytoparasites that cause estimated global annual losses of over USD 100 billion in crop yield. However, the use of registered chemical nematicides has been decreasing steadily due to concerns about their non-specific mechanism of action, potential toxicity, and the risk of environmental harm [17]. These challenges highlight the need for environmentally friendly and cost-effective alternatives to chemical nematicides, which should not harm vertebrates, crops, or other non-target organisms. Finding such alternatives has become an urgent necessity to ensure sustainable and eco-friendly pest management practices. Biocontrol strategies, such as the use of nematophagous fungi, secondary metabolites of microorganisms, and metal nanoparticles, have shown promise as an effective and less toxic alternative to chemical nematicides. Among these, nano metal particles like Ag, Au, and Cu have been extensively studied and proven to successfully control the growth of nematodes [18,19,20,21]. Building upon the knowledge from these previous studies, the present study is aimed at assessing the nematicidal potential of newly synthesized naringenin-nano Ag and Au conjugates against *Meloidogyne* sp. These nanoconjugates hold the potential to serve as a better alternative for controlling the damage caused by nematodes.

Therefore, in the current study, the local isolate of *Neurospora* sp. was explored for the production of naringinase, and citrus peel waste served as the source of naringin, which was effectively transformed into naringenin. The biotransformed naringenin and naringenin-nano Ag and Au were fabricated and tested for their antimicrobial activities against the bacterial pathogens and nematodes. Overall, this research offers a sustainable and eco-friendly solution to utilize fruit peels, reducing the waste and unlocking the potential of naringenin nanoconjugates for their enhanced antimicrobial activity.

## 2. Materials and Methods

### 2.1. Isolation of Neurospora Isolates

*Neurospora* sp. was isolated from agricultural soil. The citrus plant residue samples were collected from citrus fields. The standard serial dilution and spread plate techniques were employed to isolate the *Neurospora* sp. To a slightly modified Vogel’s Minimal Medium Agar-agar–3 g, K_2_HPO_4_–0.7 g, KH_2_PO_4_–0.2 g, MgSO_4_–0.01 g, trace element solution (FeSO_4_–0.02 g, ZnSO_4_–0.02 g, CuSO_4_–0.002 g, MnSO_4_–0.005 g, H_3_BO_3_–0.005 g)–1 mL Vogel’s trace salts (Na_2_EDTA 2H_2_O–0.1 g, NiCl_2_ 6H_2_O–0.002 g, CoCl_2_–0.002 g)–1 mL, sucrose (0.01 g) was added as a carbon source for initial growth, with 1% naringin serving as the sole carbon source. The pH was adjusted to 6.0. These components were mixed in distilled water, and then adjusted to a volume of 100 mL, and the medium was sterilized by autoclaving. The plates, inoculated with soil dilutions, were incubated at 28 °C for 48 h to allow the fungal growth. After incubation, selected fungi were transferred and maintained on Vogel’s Minimal Medium slants for further investigation and study [22].

### 2.2. Fungus Slide Culture and Staining and Molecular DNA Sequencing Identification

A small square (1 cm × 1 cm) of agar block was placed on a sterile glass slide. The agar block was inoculated with a small amount of the *Neurospora* sp. spore on each of the four sides of the block. Subsequently, a heat-sterilized coverslip was laid over the block, and it was incubated at room temperature for 48 h. Once the desired growth was observed, the coverslips were removed, and a few drops of lactophenol cotton blue stain were added. The specimen was then observed under a Zeiss AXIO Imager 2 microscope (45×) from Germany to take digital images of *Neurospora* sp. fungi stained with lactophenol cotton blue for morphological characterization [23].

The genomic DNA of the provided culture, which was obtained in its purest form, was amplified successfully to the ITS region of rDNA using fungal universal primers ITS_5 and ITS_4. A sequencing PCR was carried out using the ABI-BigDye^TM^ Terminator v3.1 Cycle Sequencing Kit. The raw sequence data obtained from the Seqstudio automated DNA sequencer were manually edited carefully to remove any inconsistencies. After editing, the obtained sequence was queried against rRNA/ITS databases that contained ITS sequences of fungi prototype and other reference materials to determine its lineage [24,25].

### 2.3. Screening of Naringinase-Producing Neurospora Strain

The screening of the potent naringinase producer, *Neurospora*, was carried out using the double-screen plate assay method. *Neurospora* sp. was inoculated onto a modified Davis minimal medium consisting of K₂HPO₄ (7.0 g/L), KH₂PO₄ (2.0 g/L), (NH_4_)₂SO₄ (1.0 g/L), yeast extract (1.0 g/L), MgSO₄ (0.10 g/L), sodium citrate (0.50 g/L), and agar-agar (30.0 g/L). For the enzyme screening of β-D-glucosidase and α-L-rhamnosidase, synthetic substrates, *p*-nitrophenyl-β-D-glucopyranoside (pNPG) (Sigma Aldrich Powai, Mumbai) and *p*-nitrophenyl-α-L-rhamnoside, were incorporated into the medium at a concentration of 10 mg/100 mL each [14]. In this assay, the isolates of *Neurospora* were spotted onto agar plates containing naringin and then incubated at 28 °C for 48 h. After the incubation period, color changes surrounding the growth of fungi were observed and measured. To further investigate its potential, the isolate’s ability to utilize complex substrates as a carbon source was examined. This was accomplished by incorporating orange peels into the agar medium, creating an environment where the isolate could grow and access naringin from the peels. The same steps used in the initial plate assay were then repeated with this orange peel agar medium.

### 2.4. Production of Naringinase Enzyme by Isolated Neurospora sp.

In this study, a 48 h old culture of *Neurospora* sp. was inoculated in a broth containing naringin and orange peels as a source of the substrate. The culture was then shaken for 24 to 144 h to allow the production of naringinase. After the incubation period, the cell mass was removed from the broth using filtration, leaving behind the filtrate containing the crude enzyme. This filtrate was used to assess naringinase production through enzyme assay and naringenin production using thin-layer chromatography (TLC). To examine the naringinase production potential of *Neurospora* sp. over time, 1 g of fermented peels was extracted in acetate buffer (pH 6.0 ± 0.2) at different intervals, i.e., 24, 48, 72, 96, and 120 h. The mixture of naringinase enzyme consisted of 0.2 mL of crude enzyme extract and 0.3 mL of 0.1 M sodium acetate buffer, (pH 6), containing 0.1% naringin substrate. The mixture was incubated at 40 °C for a period of 30 min. The measurement of naringenin concentration in the reaction mixture was determined by the Konrad Habelt and Fritz Pittner spectrophotometric method (1983) [26]. It is defined as the amount of enzyme that hydrolyzes one micromole of naringin per minute, and the results were subjected to statistical analysis. Additionally, a zero-minute incubation mixture of substrate and crude enzyme was used as a control. The entire experiment was performed in triplicate, the results were reported as mean values ± standard deviation, and data analysis was carried out using Microsoft Excel 2000.

### 2.5. Biotransformation of Naringin to Naringenin by Neurospora sp.

The active growth-phase culture of *Neurospora* was centrifuged at 7514 g to separate the fungal biomass and supernatant. The 10 g fungal biomass of *Neurospora* growth was suspended in 100 mL (*w*/*v*) of saline (0.85% NaCl in 100 mL sterile distilled water) and subsequently subjected to ultrasonication at 30 kHz for 5 min. The sonicated material was then filtered through Whatman filter paper No. 1, and the filtrate was collected. Ten microliters of the filtrate was added to the agar well in the naringin agar plate and incubated for 30 min, and after the incubation period, the color change was observed. Additionally, 10 µL of filtrate obtained from *Aspergillus brasiliensis* MTCC1344 was considered as a positive control, as the *Aspergillus brasiliensis* was previously reported for the production of nariginase [27]. After 48 h of incubation, the yellow color zone around the well was measured for both cultures’ filtrates, i.e., *Neurospora* sp. and *Aspergillus brasiliensis*.

### 2.6. Identification and Confirmation of Biotransformed Naringenin by Thin Layer Chromatography (TLC), ^1^H NMR and UV-Vis Spectroscopy

#### 2.6.1. Thin Layer Chromatography (TLC) and ^1^H NMR

An aliquot of the extracted biotransformed product in ethyl acetate was spotted on a TLC plate coated with Silica Gel 60 F254. The TLC plate was then developed using a solvent system consisting of acetone, chloroform, and water in the ratio of 80:20:4.8 [28,29]. To confirm the biotransformation, standard naringin (obtained from Sigma-Aldrich Powai, Mumbai) and product naringenin (also from Sigma-Aldrich Powai, Mumbai) were also spotted on the TLC plate. After the TLC plate development, it was visualized under long UV light in the range of 315–400 nm, and the R_f_ values were compared to confirm the biotransformation. ^1^H nuclear magnetic resonance (^1^H NMR) (BrukerMSL-300) analysis was performed on the purified biotransformed product dissolved in deuterated dimethyl sulfoxide (DMSO-d₆) solvent for structural confirmation of the naringinase product.

#### 2.6.2. Habelt and Pittner Spectrophotometric Method

The extracted naringenin product was not only confirmed but also quantitatively measured using the spectrophotometric method. For this purpose, 100 μL aliquots containing the crude biotransformed product were mixed with 3 mL of 4 N NaOH and then incubated at ambient temperature for 20 min to develop a yellow color. To quantify the naringenin, the absorbance of the samples was measured using a UV-1800 UV-VIS spectrophotometer at specific wavelengths. The absorbance was recorded at 375 nm for naringin and prunin and specifically at 310 nm for naringenin [26].

### 2.7. Preparation of Naringenin-Nano Silver (Ag) and Gold (Au) Conjugates and Their Characterization

The extracted crude products were first lyophilized to obtain a dry and concentrated form and subsequently the purified naringenin. The purified 100 mg of naringenin was dissolved in 50 mL of 50% ethanol. Additionally, a standard solution of naringenin was prepared at a concentration of 0.5 mg/mL in 50% ethanol. To examine the formation of naringenin-nano silver conjugate, 2.5 mL of a 0.01 µg/mL solution of silver nitrate (AgNO_3_) and gold chloride (AuCl_3_) were added to both the test naringenin and the standard naringenin solutions. The mixtures were thoroughly mixed to ensure homogeneity. After mixing, the solutions were exposed to sunlight for 5 min. The formation of the naringenin-nano Ag and Au conjugates was assessed by monitoring the absorption properties in the solution state using spectrophotometric scanning in the range of 200–1100 nm (Shimadzu 1605 Spectrophotometer, Tokyo, Japan).

A transmission electron microscope, the Talos F200i (S) TEM(HRTEM-200KV) model made by Thermo Fisher Scientific, with resolution 0.10 nm, point resolution of less than 0.25 nm and magnification 50× to 1M×, was used for the analysis of naringenin and naringenin-nano silver and gold nanoconjugates to gain knowledge on the shape, size and surface structural features. Samples for TEM analysis were recorded on carbon-coated copper grids. Energy Dispersive X-ray Analysis EDXA (Bruker X Flash 6 30 EDS) was used for elemental composition analysis of the nano materials. The interactions between the naringenin and nano metals were studied by FT-IR 8400 (Shimadzu FT-IR 8400, Tokyo, Japan) over a wide spectral range, which is often from the mid-infrared to near-infrared (400–4000 cm^−1^) with the resolution limits of 0.5 cm^−1^–0.01 cm^−1^. The sample preparation was carried out using KBr pelletization. 

### 2.8. Combinational Antibacterial Studies of Naringenin and Naringenin-Nano Ag and Au Conjugates

The combinational antibacterial activity of naringenin and naringenin nanoconjugates was evaluated using the well diffusion assay against four bacterial pathogens: *Staphylococcus aureus* (NCIM 2492), *Bacillus subtilis*, *Escherichia coli* (NCIM2139), and *Pseudomonas aeruginosa* (NCIM 2200). In this assay, the bacterial inoculum of ~10^5^ CFU/mL was spread evenly on the agar plates. Wells were made in the agar, and into these wells, 50 µL of naringenin-nano Ag and Au conjugates and standard naringenin (10 µg/mL) were added. The plates were then incubated at 37 °C for 24 h. After the incubation period, the diameter of the inhibition zones around each well was measured. And MIC and MBC of conjugates were also determined as per standard method [30,31]. The experiments were performed in triplicate, and the data were presented as mean ± SD. To assess the synergistic or enhanced effect of naringenin-nano Ag and Au conjugates, the activity enhancement was calculated using the formula shown below [32,33].
(1)$$Increases\;fold\;activity\;Ag=\frac{[{\left(Zone\;of\;inhibition\;of\;naringenin-nano\;Ag\;conjugate\right)}^{2}-{\left(Zone\;of\;inhibition\;of\;standard\;naringenin\right)}^{2}}{{\left(Zone\;of\;inhibition\;of\;standard\;naringenin\right)}^{2}}$$

### 2.9. In Vitro Nematicidal Activity

In this experiment, the lyophilized naringenin-Ag and Au nanoconjugates were tested for their nematicidal activity against *Meloidogyne* sp. Different concentrations of naringenin, naringenin-nano Ag, and Au conjugates were prepared in the range of 10–100 ppm using DMSO as a solvent. Copper sulphate (0.1 M) was used as a positive control, known for its nematicidal activity, while DMSO served as the negative control. The nematode suspension, containing 50 juveniles, was added to plastic wells in triplicate and kept standing at 24 °C in the dark. After a 48 h incubation period, the numbers of dead and live nematodes were counted to determine the percentage of mortality caused by each treatment [34]. To confirm the mortality, the dead nematodes were transferred to fresh distilled water, and their movement was observed for 24 h. Lack of movement during this observation period would confirm their mortality.

## 3. Results

### 3.1. Identification of the Naringinase-Producing Fungus Isolate

The slide culture technique is a valuable method for gaining insights into the morphology and characteristics of these *Neurospora* sp. The staining with lactophenol cotton blue of *Neurospora* species revealed their distinctive conidiophores and conidia. By employing the slide culture technique, one can conduct a detailed examination of the conidiophore structure, size, and arrangement, as well as the morphology of the conidia (Figure 1b). Septate, smooth walled, hyaline, eguttulate, 5.25–8.93 µm, Conidiophores oblique, 59.7–61.4 × 112.4–130.4 µm, mycelial (hyphae) breaks and arthrospores produced in branched chain; arthrospores variable in shape and size, both ends widely truncate, smooth walled, hyaline, 9.79–25.17 × 4.4–7.4 µm; asexual stage absent. This indicated that the isolated fungi belonged to the genus *Neurospora*.

Sequence analysis using the NCBI accession number MT102855.1 for the *Neurospora crassa* strain indicated that the tested fungal strain had 100% sequence similarity with *Neurospora crassa* (Figure 2).

### 3.2. Neurospora sp. Mediated Biotransformation of Naringin to Naringenin

Screening of *Neurospora* sp. by the naringinase double-screen plate assay method showed that the yellow zone surrounding the colony/growth indicated naringinase production [35]. These findings suggest that *Neurospora* sp. possesses a high potential for transforming naringin (Figure 3a). Furthermore, the biotransformation of naringin by *Neurospora* was confirmed through both enzyme assay and spectrophotometric analysis, following the method previously described by Habelt and Pittner in 1983 [26]. The spectrophotometric analysis displayed a characteristic peak at 310 nm, which corresponded to NAR23, confirming the presence of naringenin (Figure 3b). Additionally, the substrate naringin exhibited a peak at 375 nm, further corroborating the conversion of naringin into naringenin. These results provide compelling evidence for the successful biotransformation of naringin by *Neurospora* sp.

### 3.3. Thin Layer Chromatography and NMR Analysis

The chromatography results under short UV rays demonstrated the formation of a new product, evident from the bands observed at different R_f_ values compared to the substrate naringin band (Figure 3c). The naringenin, produced through the enzymatic activity of *Neurospora* naringinase, was further characterized by ^1^H NMR and compared with standard NAR [36]. A comparison of the spectra revealed that the peaks of the biotransformed product matched with those of the standard naringenin, with some additional peaks that could be possibly attributed to impurities. The observed peaks were as follows: d = 7.28 (d, *J* = 8.7, 1H), 6.77 (d, *J* = 8.5 Hz, 1 H), 6.04 (d, *J* = 2.4 Hz, 1H), 5.36–5.48 (m, 1H), 4.98–5.11 (m, 1 H), 3.40–3.47 (m, 2 H), 1.12 ppm (d, *J* = 6.1 Hz, 1 H). The ^1^H NMR spectrum shows two hydrogen atom signal shift values (Table 1) in A ring and three B ring of naringenin (Figure 3e) [37,38]. For comparison, the HPLC chromatograms of the standard substrates, naringin and naringinase, and the biotransformed product (naringenin) are shown in Appendix A. 

### 3.4. Production of Naringinase Enzyme by Neurospora sp. Utilizing Orange Peels

The study focused on evaluating the time-dependent production of naringinase enzyme by *Neurospora* sp. using orange peels as the substrate. The results indicated a progressive increase in enzyme production with longer incubation times. After 96 h of incubation, the maximum enzyme was recorded at 389 IU/mL (Figure 4, Appendix A). The enzyme activity, specifically of naringinase, was observed to increase over time. For instance, at 48 h of incubation, the enzyme activity showed a 3.73-fold increase compared to the initial level. This increase further reached 9.5-fold at 72 h of incubation and continued to consistently rise up to 96 h. However, beyond 96 h, there was a slight decline in the enzyme activity (Figure 4).

### 3.5. Synthesis of Naringenin-Nano Ag and Au Conjugates

The naringenin-nano Ag and Au conjugates were synthesized under sunlight. When naringenin and metal substrates interacted with each other, sunlight accelerated the reduction reaction, leading to the formation of naringenin-nano Ag and Au conjugates. The reaction mixtures underwent a visible color change, turning from colorless to yellowish-brown for Ag nano-conjugates and from light yellow to ruby red for Au nano-conjugates (Figure 5b). The naringenin-nano Ag conjugate exhibited a characteristic absorption peak at a wavelength of 420 nm, while the Au nanoconjugate showed an absorption peak at a wavelength of 590 nm, which indicated the successful formation of the respective Ag and Au nanoconjugates with naringenin (Figure 5a).

### 3.6. Characterization of Nanoconjugates of Ag and Au with Naringenin

The structure and morphology of the synthesized nanoconjugates were analyzed using transmission electron microscopy (TEM). The TEM micrograph of naringenin-nano Ag and Au conjugates revealed uniformly distributed and triangular, spherical-shaped Ag particles (Figure 6b) and spherical and polygonal particles for the Au conjugate (Figure 6a). The average particle size of Ag and Au naringenin nanoconjugates was 80.82 ± 3.2 nm and 65.25 ± 3.9 nm, respectively, and it can be stated that these were biologically active nanoconjugates. The zeta potential values of Ag nanoconjugates (−25.3 mV) and Au nanoconjugates (−22.9 mV) indicate the stability of the nanoconjugates due to electrostatic repulsion between particles. It helps to avoid aggregation behavior of nanoconjugates. The average particle size values deduced from the diffused light scattering (DLS) studies are also shown in Appendix A, supporting the argument that the nanoconjugates are stable without aggregation. Likewise, the zeta potential distribution traces are also shown for the nanoconjugates of Ag and Au with naringenin (Appendix A). In addition to the UV-Vis studies shown in Figure 5, to further ascertain the presence of elemental Ag and Au in the synthesized nanoconjugates, an energy-dispersive X-ray spectroscopy (EDXA) analysis was conducted. The EDXA profile of naringenin-nano Ag and Au showed characteristic signals authenticating the presence of Au and Ag in the naringenin nanoconjugates (Figure 6c,d). The interaction and role of naringenin in the synthesis of nano Ag and Au were investigated using Fourier transform infrared spectroscopy (FT-IR). The FT-IR analysis of naringenin showed characteristic peaks at 3055 cm^−1^ (corresponding to -OH stretching vibration), 1595 cm^−1^ (C=C stretching vibration), and 1175 cm^−1^ (C-O stretching vibration). In comparison, the synthesized naringenin-nano Ag conjugate showed slight shifts in bands at 3289 cm^−1^ (-OH), 1627 cm^−1^ (C=C), and 1181 cm^−1^ (C-O), while the naringenin-nano Au conjugate exhibited peak shifts at 3297 cm^−1^ (-OH), 1627 cm^−1^ (C=C), and 1181 cm^−1^ (C-O). Notably, the peak shifts were approximately the same for both naringenin-nano Ag and naringenin-nano Au conjugates (Appendix A).

### 3.7. Antibacterial Activity of Synthesized Naringenin-Nano Ag and Au Conjugates

When tested as such, naringenin exhibited moderate antibacterial activity against both Gram-positive and Gram-negative bacteria. The naringenin-nano Au conjugate also displayed similarly moderate activity against the tested bacterial strains, with an increase of 0.12–0.28 fold in antibacterial activity compared to standalone naringenin (Table 2). In contrast, the Ag nanoconjugate showed significantly enhanced antibacterial activity against both Gram-positive and Gram-negative bacteria compared to standalone naringenin (Figure 7). The naringenin-nano Ag conjugate demonstrated a remarkable 3.7-fold increase in activity against *S. aureus* and an impressive 8.2-fold increase against *P. aeruginosa.*

Tests related to the determination of the minimum inhibitory concentration (MIC) and the minimum bactericidal concentration (MBC) of the nanoconjugates were conducted. The results revealed an MIC of 0.312 mg/mL and an MBC of 0.625 mg/mL for both gram-positive and gram-negative bacteria, i.e., *S. aureus ATCC 6538* and *E. coli K12.* These results agree well with the previous report on silver nanomaterial. Parvekar et al. 2020 [30]. These finding have been summarized in the Electronic Appendix A. 

### 3.8. Nematicidal Activity of Biotransformed Product Naringenin and the Naringenin-Nano Ag and Au Conjugates

Until now, naringenin itself has not been reported for its nematicidal activity. However, Goyal et al. reported on the nematicidal activity of certain compounds, including oils and polyphenols, extracted from citrus peels [39]. In our current study, we aimed at evaluating the nematicidal potential of naringin, naringenin, and nanoconjugates of naringenin with Ag and Au against *Meloidogyne* sp. The results of the bioassay revealed that naringin did not exhibit any activity against the tested nematodes. However, after the biotransformation of naringin to naringenin using *Neurospora* naringinase, naringenin displayed nematicidal activity with an LC50 value of 88.72 mg/L (Table 3). Moreover, the synthesized naringenin Ag and Au nanoconjugates exhibited nematicidal activity, with the Ag nanoconjugate showing an LC50 value of 46.23 mg/L (47.90% lower compared to naringenin) and the Au nanoconjugate showing an LC50 value of 61.43 mg/L (30.75% lower compared to NAR). Notably, the nematicidal activity of both nanoconjugates surpassed that of standalone NAR and was comparable to that of the standard nematicide, CuSO_4_. These findings highlight the superior nematicidal potential of the nanoconjugates of Ag and Au with naringenin in comparison to naringenin and indicate their promising role as nematicides.

## 4. Discussion

Naringin undergoes enzymatic hydrolysis, catalyzed by naringinase, to convert it into the bioactive compound NAR. Several species of microorganisms have been reported for this enzymatic conversion of naringin to naringenin [12,40]. However, much focus has been placed on fungal species for the production of the naringinase enzyme and its use in converting naringin to naringenin. Among the fungal genera, *Aspergillus* has been extensively explored for naringinase production, with approximately nine species of Aspergillus and Penicillium and *Rhizopus* sp. reported for this purpose [41]. Although, the fungus is a good source for naringinase production, a significant drawback is that the time required for growth and production of biomolecules could be days or sometimes months. In contrast, *Neurospora* sp. overcomes these drawbacks with additional advantages. Its well-known genome is easy to manipulate for enzyme production, and it has a high growth rate—one of the important characteristics for large-scale industrial production. Moreover, it is considered safe in the food industry, as, to date, there is no evidence of *Neurospora* sp. being harmful or pathogenic to humans or plants [9]. In the present study, we aimed at overcoming this obstacle by screening the non-pathogenic *Neurospora* sp. for the production of naringinase and subsequently bio transforming naringin to naringenin. The selection of *Neurospora* was based on its origin from agricultural soil in a citrus field. The isolate exhibited remarkable naringinase-producing capability, in conditions both when grown in well-defined medium supplemented with naringin and while using citrus peels as a carbon source. The quantification of the enzyme produced by *Neurospora* sp. reached 389 IU/mL after a 96 h incubation period. The production of naringinase and the maximum conversion of naringin to NAR were observed to be time-dependent, with an optimal time requirement of 96 h. These results outperformed earlier reports of bacterial naringinase production, such as with *Serratia* sp. and *Bacillus methylotrophicus*, with reported activities of 9.2 and 12.05 IU/L, respectively [42]. Furthermore, when our results were compared with the production of naringinase by *Aspergillus oryzae* JMU316 (408.28 IU/mL), the values were comparable or slightly lower [43]. These results demonstrate the significant naringinase-producing capacity of *Neurospora* sp., making it a promising candidate for large-scale production of naringenin through biotransformation of naringin. The pharmacological activities of NAR that have garnered significant interest among researchers include antioxidant, antibacterial, and anticancer activities [44,45]. Furthermore, the fabrication of naringenin with metal nanoparticles and the formation of nanoconjugates improved its bioactivities. The *Aspergillus* sp. mediated biotransformed NAR functionalized with nano Ag enhanced the antimicrobial potential by 4.5 to 13.6 fold over the standalone activity of naringenin against common bacterial pathogens including both gram-positive and gram-negative bacteria [14]. Similarly, against Acanthmoeba cells, the naringenin-nano Ag conjugate significantly reduced the LC50 values by 50.56%, as compared to naringenin. Borase et al. in 2013 demonstrated similar amoebicidal activity of phytosynthesized Ag nanomaterials [46]. Among various metal nanoparticles, Ag and Au have garnered much attention for their antimicrobial activities. For instance, Ag nanoparticles derived from fungi have demonstrated remarkable activity against both *E. coli* and *S. aureus* [47]. Regarding Au nanoparticles, there have been mixed reports on their antibacterial activity. Some studies highlight potent antibacterial effects, like the research by Li et al. in 2014, which reported strong antibacterial activity of Au nanoparticles against multidrug-resistant *S. aureus* [48]. On the other hand, Skladanowski et al. indicated that even at the highest concentration (182 µg/mL), Au nanoparticles were ineffective against gram-negative bacteria [49]. In our study, we observed similar results for the naringenin-nano Ag conjugate, which displayed remarkable antibacterial activity compared to naringenin against both gram-positive and gram-negative bacteria and moderate activity for Au nanoconjugates against the tested bacterial pathogens. The synthesized Ag nanoconjugate’s results align with the findings of Gakiya et al. [16], wherein it was reported that an Ag nanoparticle peptide conjugate exhibited greater antibacterial activity than the peptide alone. Similarly, Ag conjugates of Cephradine and Vildagliptin were found to be more effective against various bacteria, including *P. aeruginosa*, *K. pneumoniae*, *E. coli K1*, *S. pyogenes*, methicillin-resistant *S. aureus* (MRSA) and *B. cereus* [50,51]. In addition to these findings, it was demonstrated that silver nanoconjugates of Nisin (Ag-Nisin) exhibited enhanced activity against biofilm formations and other microbes, such as *S. aureus* and *E. coli*, compared to Nisin alone [15]. These results collectively suggest that synthesizing metal nanoconjugates, particularly Ag and Au nanoconjugates, can significantly boost the antimicrobial potential of natural compounds like NAR and other peptides, making them promising candidates for combating bacterial infections and biofilm formations. In addition to the antibacterial activity, we further explored the nematicidal activity of both of the nanoconjugates using *Meloidogyne*. The results revealed that naringenin on its own showed marginal nematicidal activity. Nevertheless, the nematicidal potential of biotransformed naringenin was significantly enhanced when it was functionalized and with the formation of nano-Ag and Au conjugates. To the best on our limited knowledge, naringin and naringenin have not yet been reported for their activity against nematodes. This is the first time we explored the potential of naringenin as a nematicidal agent. However, on the other hand, the Ag particles have been extensively studied for nematicidal activity using the model organism *C. elegans* as well as field isolates of nematodes. The Ag particles, synthesized and fabricated by using different chemical and biological routes, exhibited significant mortality at lower lethal doses against the tested nematodes. In 2014, Cromwell et al. conducted a study in which they tested the chemically synthesized Ag nanoparticles (with sodium borohydride and 0.2% starch as a stabilizer) against *Meloidogyne incognita*. The results of the study evidenced that the nematodes in soil were reduced by 92% and 82% after 4 and 2 days of exposure to 150 mg/mL of Ag nanoparticles [19]. Similarly, Mahmoud et al. (2016) evaluated the *Bacillus pumilus* mediated Ag nanoparticles against *Panagrellus redivivus*. The 150 and 200 µg/mL concentrations of Ag nanoparticles showed significant mortality with 80 and 91% death rates, respectively, after 48 h exposure [18]. The naringenin-nano Ag also showed excellent nematicidal activity against *Meloidogyne* sp. with a nematode mortality rate of 50% at 46.23 mg/L. Additionally, the naringenin-nano Au also inhibited 50% of nematodes at 61.43 mg/L, which is noteworthy. The superior activity of naringenin nanoconjugates could be due to the cumulative action of naringenin and Ag or Au, which help to inhibit the nematodes at lower doses. The interaction of Ag nanoparticles with nematodes is not specific but associated with oxidative stress-related PMK-1 P38 MAPK activation [52]. Nonetheless, further studies are required to explore the mechanism of action of naringenin alone and its nanoconjugates with plasmonic nanomaterials like Ag and Au against nematodes.

## 5. Conclusions

The enzymatic biotransformation of naringin to bioactive naringenin (NAR) by using the non-pathogenic fungus *Neurospora* sp. as a source of naringinase enzyme is a sustainable alternative to the existing methodologies. Additionally, *Neurospora* can utilize citrus peels as a carbon source to produce the naringinase enzyme, and consequently, naringenin from the citrus waste. It is an economically viable and fast biotransformation of naringin to naringenin. Furthermore, the biotransformed naringenin with the ability to reduce the bulk silver and gold substrates to their nano forms resulted in the formation of nanoconjugates with enhanced antimicrobial activities. The nanoconjugates showed potential biomedical applications, including antibacterial activity, and could also contribute to the development of sustainable and eco-friendly approaches to nematode control, reducing the reliance on chemical nematicides and their adverse effects on the environment and non-target organisms. Overall, the new concept of fabricating nanoconjugates of naringenin with Ag and Au nanoparticles would help to improve its pharmacological potential and nematicidal activity, providing an alternative to high-risk synthetic nematicides. 

However, there are some limitations of the study, such as the issue of toxicity of newly synthesized naringenin-nano Ag and Au conjugates, which has not yet been studied. Moreover, the species of nematode used in this study was an agricultural isolate. Studies on models as well as nematodes from human and animal species are further warranted. These additional studies would strengthen the proposed application of naringenin Ag and Au nanoconjugates for protection from bacteria as well from nematode infections.

## Figures and Tables

**Figure 1 bioengineering-11-00510-f001:**
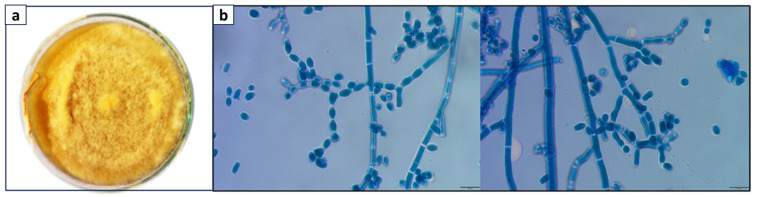
(**a**) *Neurospora* sp. active growth on agar plate. (**b**) Microscopic examination of *Neurospora* sp. stained with lactophenol cotton blue.

**Figure 2 bioengineering-11-00510-f002:**
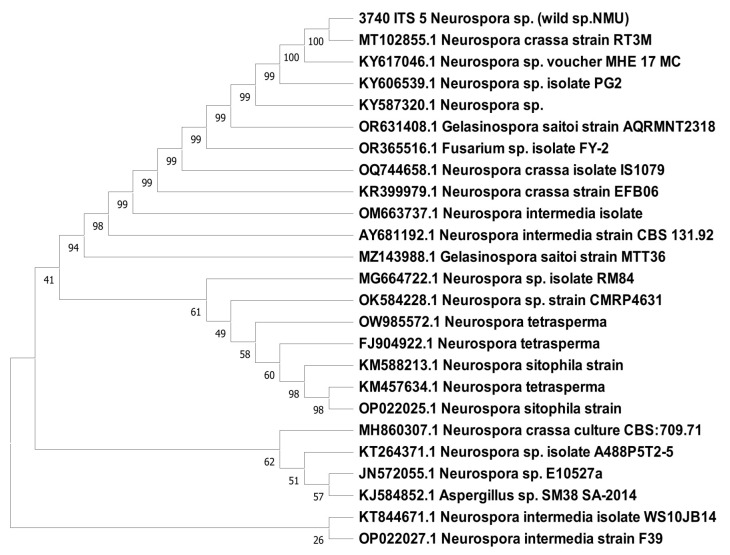
Phylogenetic tree of isolated *Neurospora crassa* identified by using ITS region of rDNA sequence (neighbor-joining method).

**Figure 3 bioengineering-11-00510-f003:**
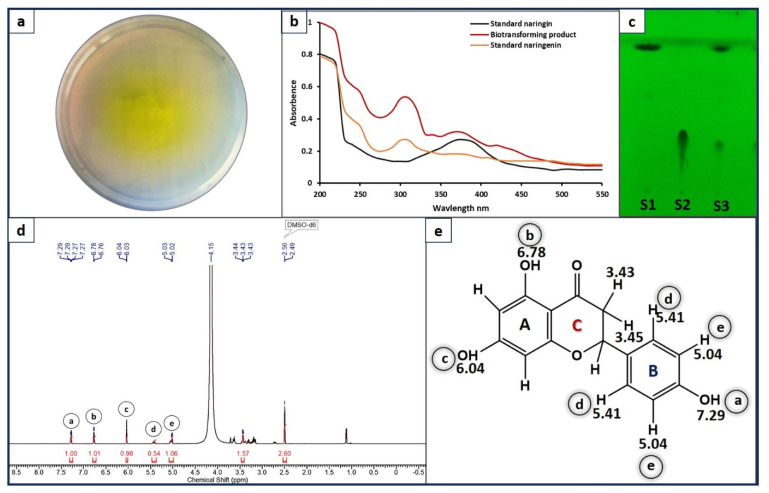
(**a**) Screening of naringinase-producing *Neurospora* sp. (**b**) Primary confirmation of transforming product by spectrophotometry (**c**) Confirmation of transforming product (naringenin) by thin layer chromatography (TLC). S1—standard naringenin, S2—standard naringin, S3—biotransformed product. (**d**) ^1^H NMR spectrum of pure biotransformed product (naringenin). (**e**) Structure of the isolated compound (naringenin, NAR) as deduced from NMR analysis.

**Figure 4 bioengineering-11-00510-f004:**
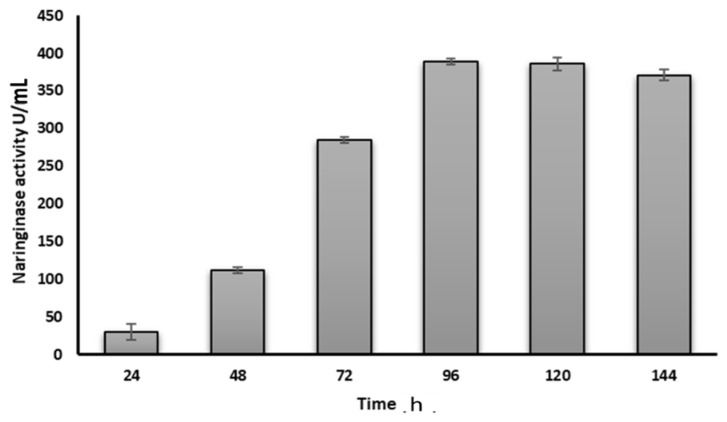
Variation of *naringinase* activity (transformation of naringin) with incubation time.

**Figure 5 bioengineering-11-00510-f005:**
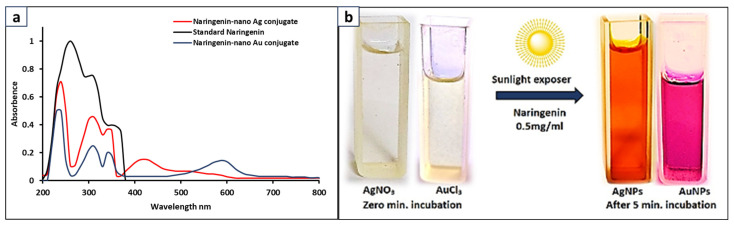
(**a**) UV-Vis spectrum confirming the formation of naringenin-nano Ag and Au conjugates vs. naringenin. (**b**) Visible color change under sunlight for formation of Ag and Au nanoconjugates with naringenin.

**Figure 6 bioengineering-11-00510-f006:**
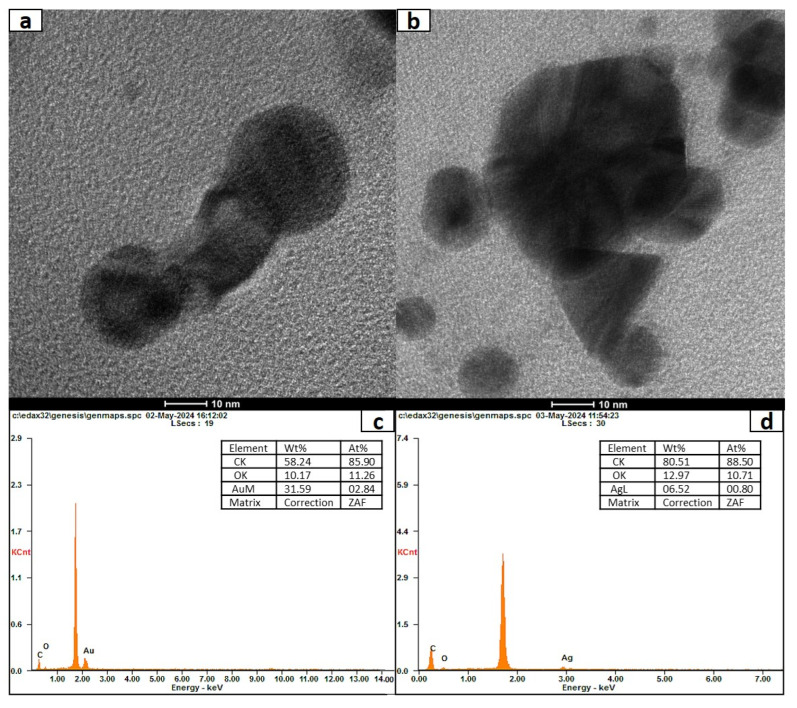
(**a**) FE-TEM naringenin-nano Au conjugate. (**b**) FE-TEM naringenin-nano Ag conjugate. Energy dispersive X-ray analysis (EDXA) plots of (**c**) naringenin-nano Au conjugate (Wt. % of Au is 31.59) and (**d**) naringenin-nano Ag conjugate (Wt. % of Ag is 6.52).

**Figure 7 bioengineering-11-00510-f007:**
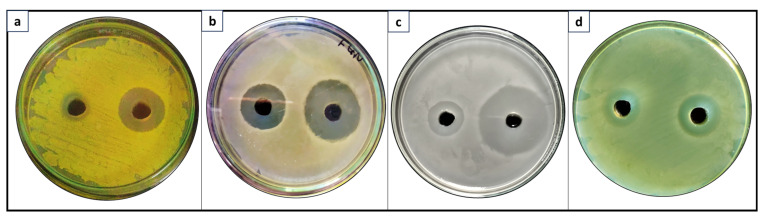
Representative antibacterial agar plate assays of the biotransformed product (left) and its nanoconjugate (right) in each agar plate carried out versus (**a**) *Staphylococcus aureus* (**b**) *Bacillus subtilis* (**c**) *Escherichia coli* and (**d**) *Pseudomonas aeruginosa*.

**Table 1 bioengineering-11-00510-t001:** ^1^H NMR chemical shift values (ppm) of naringenin (NAR).

Peak	Shift (ppm)
a	7.29
b	6.78
c	6.04
d	5.41
e	5.04

**Table 2 bioengineering-11-00510-t002:** Antimicrobial activities of naringenin, naringenin-nano Au conjugate, and naringenin-nano Ag conjugate against microorganisms.

Tested Microorganism	*S. aureus*	*B. subtilis*	*E. coli*	*P. aeruginosa*
**ZOI for naringenin (in mm)**	6.8 ± 0.17	6.6 ± 0.31	6.4 ± 0.19	4.7 ± 0.24
**ZOI for naringenin-nano Au conjugate (in mm)**	7.2 ± 0.26	7.0 ± 0.38	7.1 ± 0.29	5.0 ± 0.25
**ZOI for naringenin–Ag nanoconjugate (in mm)**	14.8 ± 0.35	12.8 ± 0.28	14.2 ± 0.23	14.3 ± 0.21
**ZOI for AgNO_3_ (in mm)**	6.2	6.4	7.3	5.9
**ZOI for AuCl_3_ (in mm)**	Nil	Nil	Nil	Nil
**Fold increase in antimicrobial activity** **(naringenin–Au nanoconjugate)**	0.126	0.124	0.282	0.131
**Fold increase in antimicrobial** **activity (naringenin–Ag nanoconjugate)**	3.737	2.761	3.922	8.251

**Note:** Experiments were carried out in triplicate, and the mean and standard deviation (SD) values were calculated. Data represented here are the values of mean ± SD.

**Table 3 bioengineering-11-00510-t003:** Nematicidal activity of naringenin compared to naringenin-Ag and naringenin-Au nanoconjugates.

Test Products	LC50 ± SD (mg L^−1^)	95% Fiducial Limits	LC90 ± SD (mg L^−1^)	95% Fiducial Limits	Regression Equation
**Naringenin**	88.72 ± 2.53	83.71–93.71	186.01 ± 6.21	174.98–199.62	Y = 4.78 + 0.320 X
**Std. (CuSO_4_)**	41.03 ± 2.51	107.35–2.89	102.06 ± 113.48	35.78–45.69	Y = 16.0 + 0.242 X
**AgNO_3_**	83.39 ±2.41	78.58–88.09	174.51 ± 5.35	174.94–186.14	Y = 4.02+ 0.250 X
**AuCl_3_**	120.62 ± 2.56	115–125	205.34 ± 6.26	194.18–219.27	Y = 3.89 + 0.242 X
**Naringenin–Au nanoconjugate**	61.43 ± 2.17	56.99–65.55	132.33 ± 3.33	126.25–139.42	Y = 12.1 + 0.257 X
**Naringenin–Ag nanoconjugate**	46.23 ± 2.55	40.89–50.99	118.39 ± 3.17	112.60–125.14	Y = 13.8 + 0.248 X

**Note:** Experiments were carried out in triplicate, and the mean and standard deviation (SD) were calculated. Y, mortality rate (significant at *p* < 0.05 level); X, concentration (significant at *p* < 0.05 level); **LC50**, lethal concentration that kills 50% of the exposed nematodes; **LC90,** lethal concentration that kills 90% of the exposed nematodes. Data represented here are the values of mean ± SD.

## Data Availability

No datasets were generated in present study Except Figure 4, raw data available with author, all figures attached with MS are original.

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
