# Peer review of "Neurospora sp. Mediated Synthesis of Naringenin for the Production of Bioactive Nanomaterials"

_bioengineering, 2024, doi:10.3390/bioengineering11050510_

Round 1

Reviewer 1 Report

Comments and Suggestions for Authors

The article “Neurospora sp. mediated Synthesis of Naringenin for the production of Bioactive Nanomaterials” by Jitendra D Salunkhe et al reports about the enzymatic biotransformation of naringin to bioactive naringenin (NAR) by using the non-pathogenic fungi i.e. Neurospora sp. The authors investigate the potential to utilize citrus peels as a carbon source for economical and rapid biotransformation of naringin to naringenin. They showed that biotransformed naringenin can reduce bulk silver and gold substrates to nanoconjugates and tested its antimicrobial and antinematicidal activities. They applied a range of biophysical characterization techniques and activity assays to analyze the produced nanoconjugates. The article is well written and understandable for the reader. However, the quality of the presentation of the results is very poor (figures) and appears to be very rushed. In some cases, the obtained results are pasted as screenshots from the data acquisition software and are not well readable. In addition, for the activity assays controls are missing using Ag and Au nanoparticles alone to test if the observed antimicrobial and antinematicidal activities are due to the NAR nanoconjugates or the Ag and Au nanoparticles alone.

Major comments:

Line: 128: Which microscope was used (Type and Company) and which imaging mode was applied

Line 154ff: The naringinase assay is described, but the method section only mentions “the enzyme activity of the extracts was tested…”, but no details are given what kind of assay was used for this.

Line 160: Give g force value instead of rpm (depends on centrifuge and rotor type)

Line 162: Specify “saline” and the sonication parameters (frequency)

Line 182: Specify the Spectrophotometer type

Line 206: Which TEM type was used and how were the samples prepared for the TEM imaging. What kind of support was used (TEM grids?)?

Line 207ff: The sentence is not complete and the TEM and FT-IR measurements need to be explained in more details.

210ff. Control experiment with Ag and Au particles alone are missing to verify that the antimicrobial effect is attributed to the naringenin-nano conjugates.

Line 308: Ag TEM micrograph is shown in Fig 5c not 5b. The shown particles are not monodisperse. The TEM images are not suitable to judge if the particles are monodisperse, only a rough size range can be estimated based on the images. Solution scattering like dynamic light scattering would be more suitable to determine the size distribution and polydispersity index.

Line 328ff: Antibacterial activity should be compared to Ag and Au nanoparticles without naringenin to confirm that naringenin is indeed increasing the antibacterial activity.

Line 344ff: Nematicidal activity should be compared to Ag and Au nanoparticles without naringenin to confirm that naringenin is indeed increasing the nematicidal activity.

Figure 2b: If a UV-vis spectrum is used to confirm conversion of naringin to naringenin, spectra of both pure components need to be included in the presented graph as control. Like this the reader is not able to judge if the new emerging peak at 310 nm really attributes to formed naringenin.

Figure 2c: The same problem as in figure 2b, a positive control of naringenin is missing. Labeling of the TLC results need to be optimized.

Figure 2d: Peak labels (a, b, c, d, e) should be added to the graph to be consistent with figure 2e and table 1. Graphic is of poor quality (resolution, no y-axis, undefined labels in red color) and should be plotted using suitable software. The text states that the spectrum was compared with standard NAR, but the reference spectrum is (again) not shown in the graph.

Figure 4a: Tick labels on both axes are missing. Labels on x-axsis seem to be not in the right place, at least they do not align with the mentioned peaks of 420 nm and 590 nm in the text.

Figure 5: EDAX graphs seem to be screenshots from the date acquisition software. Graphs are of very poor quality and not readable like this. Peaks are not identifiable over the background. Same applies for FT-IR spectra in panel

Comments on the Quality of English Language

The article is well written and I do not have concerns regarding the quality of Englisch language.

Minor comments:

Line 42: Correct to „Nowadays, …“

Line 83: Citation 14 formatting

Line 89: Citation 17 formatting

Line 124 & 131 & 238 & 247 & 249 & 387 & 390: Neurospora sp. italic formatting

Line 139: Remove “meticulously”

Line 207: “…morphology. The…”

Line 336: S. aureus italic formatting

Author Response

We are grateful to the referee for the meticulous care in evaluating our paper. The responses to the comments of the referee are provided below for the kind evaluation of the referee. 

Manuscript ID:bioengineering-2906949

Reponse to referee’s comments:

Report of reviewer 1:

General Comments and Suggestions for Authors:The article “Neurospora sp. mediated Synthesis of Naringenin for the production of Bioactive Nanomaterials” by Jitendra D Salunkhe et al reports about the enzymatic biotransformation of naringin to bioactive naringenin (NAR) by using the non-pathogenic fungi i.e. Neurospora sp. The authors investigate the potential to utilize citrus peels as a carbon source for economical and rapid biotransformation of naringin to naringenin. They showed that biotransformed naringenin can reduce bulk silver and gold substrates to nanoconjugates and tested its antimicrobial and antinematicidal activities. They applied a range of biophysical characterization techniques and activity assays to analyze the produced nanoconjugates. The article is well written and understandable for the reader. However, the quality of the presentation of the results is very poor (figures) and appears to be very rushed. In some cases, the obtained results are pasted as screenshots from the data acquisition software and are not well readable. In addition, for the activity assays controls are missing using Ag and Au nanoparticles alone to test if the observed antimicrobial and antinematicidal activities are due to the NAR nanoconjugates or the Ag and Au nanoparticles alone.

Major comments:

Comment 1: Line: 128: Which microscope was used (Type and Company) and which imaging mode was applied.

Response 1: Accepted and included in revised manuscript.

Comment 2: Line 154ff: The naringinase assay is described, but the method section only mentions “the enzyme activity of the extracts was tested…”, but no details are given what kind of assay was used for this.

Response 2: Yes, we accepted and now added the detailed assay in revised paper.

Comment 3: Line 160: Give g force value instead of rpm (depends on centrifuge and rotor type)

Response 3: Accepted and corrected in the revised manuscript.

Comment 4:Line 162: Specify “saline” and the sonication parameters (frequency).

Response 4: Accepted and introduced in the revised paper.

Comment 5: Line 182: Specify the Spectrophotometer type.

Response 5: Accepted and corrected.

Comment 6: Line 206: Which TEM type was used and how were the samples prepared for the TEM imaging. What kind of support was used (TEM grids?)?

Response 6: Carbon coated copper grids were used. Following the suggestions, the make of the TEM and the procedure for TEM analysis were added in the experimental section.

Comment 7: Line 207ff: The sentence is not complete and the TEM and FT-IR measurements need to be explained in more details.

Response 7: Accepted and revised accordingly.

Comment 8: 210ff. Control experiment with Ag and Au particles alone are missing to verify that the antimicrobial effect is attributed to the naringenin-nano conjugates.

Response 8: Accepted and now added in the concerned table.

Comment 9: Line 308: Ag TEM micrograph is shown in Fig 5c not 5b. The shown particles are not monodisperse. The TEM images are not suitable to judge if the particles are monodisperse, only a rough size range can be estimated based on the images. Solution scattering like dynamic light scattering would be more suitable to determine the size distribution and polydispersity index.

Response 9: We accept the advice and have reanalyzed the sample and the data has been introduced in the revised paper.

Comment 10: Line 328ff: Antibacterial activity should be compared to Ag and Au nanoparticles without naringenin to confirm that naringenin is indeed increasing the antibacterial activity.

Response 10:Accepted and introduced in appropriate tables in the revised paper.

Comment 11: Line 344ff: Nematicidal activity should be compared to Ag and Au nanoparticles without naringenin to confirm that naringenin is indeed increasing the nematicidal activity.

 Response 11: Accepted and revised accordingly.

Comment 12: Figure 2b: If a UV-vis spectrum is used to confirm conversion of naringin to naringenin, spectra of both pure components need to be included in the presented graph as control. Like this the reader is not able to judge if the new emerging peak at 310 nm really attributes to formed naringenin.

Response 12: Accepted and revised accordingly.

Comment 13: Figure 2c: The same problem as in figure 2b, a positive control of naringenin is missing. Labeling of the TLC results need to be optimized.

Response 13: Accepted and revised accordingly..

Comment 14: Figure 2d: Peak labels (a, b, c, d, e) should be added to the graph to be consistent with figure 2e and table 1. Graphic is of poor quality (resolution, no y-axis, undefined labels in red color) and should be plotted using suitable software. The text states that the spectrum was compared with standard NAR, but the reference spectrum is (again) not shown in the graph.

As standard product NMR available in various journal, used it for characterization but there is no need to give standard graph of product.

Response 14: Accepted and revised accordingly.

Comment 15: igure 4a: Tick labels on both axes are missing. Labels on x-axis seem to be not in the right place, at least they do not align with the mentioned peaks of 420 nm and 590 nm in the text.

Response 15: Accepted and corrected..

Comment 16: Figure 5: EDAX graphs seem to be screenshots from the date acquisition software. Graphs are of very poor quality and not readable like this. Peaks are not identifiable over the background. Same applies for FT-IR spectra in panel.

Response 16: We added revised figure which is now more clear compared to the previous version.

Comment 17: On the Quality of English Language-the article is well written and I do not have concerns regarding the quality of Englisch language.

Response 17: We thank the referee for the appreciation.

Minor comments:

Comment 18: Line 42: Correct to „Nowadays, …“

Response 18: The correction has been made as shown in track changes mode.

Comment 19: Line 83: Citation 14 formatting

Response 19: The suggested correction was done.

Comment 20: Line 89: Citation 17 formatting

Response 20: The suggested correction was done.

Comment 21:Line 124 & 131 & 238 & 247 & 249 & 387 & 390: Neurospora sp. italic formatting.

Response 21: Following the advice, the change has been made as shown in track changes mode.

Comment 22: Line 139: Remove “meticulously”

Response 22: The suggested change was made.

Comment 23: Line 207: “…morphology. The…”

Response 23: We thank the referee in meticulously reading our paper line by line and pointing out the error. The mistake was corrected as shown in track changes mode.

Comment 24: Line 336: S. aureus italic formatting

Response 24: The change has been done.

Reviewer 2 Report

Comments and Suggestions for Authors

See document attached

Author Response

We are thankful to the referee for the critical inputs for improving the quality of the paper.  All the suggested changes were made in the revision and a point by point response to referee's comments are shown below: 

Report of reviewer 2:

General comments: Authors have presented a paper for converting Naringin to Naringenin by Neurospora sp. There are some interesting results but there is a lack of information that should be addressed. For that reason I don’t recommend this article for publication.

Major comments:

Comment 1:How can Neurospora access naringin from orange peels. That is difficult to understand and it should be clarified. If it is a standard protocol please provide the reference.

Response 1: Neurosporas sp. was well documented for its ability to grow on various wild and organic substrates. The organism was well characterised and reported for potential to produce various enzymes essentially needed to grow in organic and solid substrates i.e. amylase, cellulases, xylanases etc. For the naringin biotransformation, the required enzymes i.e. glucosidase and the organism were also documented. The related references were given below. Although there is no report on rhamnosidase, we confirmed its rhamnosidase producing potential by testing it on the minimal media containing standard synthetic substrate i.e. p-nitrophenyl-α-l-rhamnopyranoside, PNPR (Sigma Aldrich);  the liberation of p-Nitrophenol confirms the rhamnosidase potential. We added this part in description of strain in  the revised paper as advised by the referee.

References:

Shahryari Z, Fazaelipoor MH, Ghasemi Y, Lennartsson PR, Taherzadeh MJ. Amylase and Xylanase from Edible Fungus Neurospora intermedia: Production and Characterization. Molecules. 2019 Feb 17;24(4):721. doi: 10.3390/molecules24040721. PMID: 30781572; PMCID: PMC6412995.

  1. T. Yazdi, A. A. Khosravi, M. Nemati, and N. D. V. Motlagh, “Purification and characterization of two intracellular β-glucosidases from the Neurospora crassamutant cell-1,” World Journal of Microbiology & Biotechnology, vol. 19, no. 1, pp. 79–84, 2003.

Nair R.B., Lundin M., Lennartsson P.R., Taherzadeh M.J. Optimizing dilute phosphoric acid pretreatment of wheat straw in the laboratory and in a demonstration plant for ethanol and edible fungal biomass production using Neurospora intermediaJ. Chem. Technol. Biotechnol. 2017;92:1256–1265.

Lin, L., Wang, S., Li, X., He, Q., Benz, J. P., and Tian, C. (2019). STK-12 acts as a transcriptional brake to control the expression of cellulase-encoding genes in Neurospora crassa. PLoS Genet. 15:e1008510. doi: 10.1371/journal.pgen. 1008510

Salunkhe, J. D., Patil, S. V. (2023). Improved naringinase double screen plate assay: progress

towards the perfect screening. Natural Product Research, 1-4.

Comment 2: Authors evaluate the activity of bioconversion by using the extracts without considering the amount of protein and the purification of them. It is really important to perform some experiments to identify properly the presence of the enzyme. For example, Western blott or affinity chromatography.

Response 2: The common enzyme purification method we used  results in obtaining the crude enzyme and the enzyme confirmation done by the testing of the extract for glucosidase and rhamnosidase activity qualitatively using standard synthetic substrates. In addition,the crude enzyme tested showed the Naringenin formation after utilizing Naringin as substrate and the product formed was analyzed with standard product by TLC, spectrophotometry and by biochemical reaction of Habelt and Pittner (1983). Moreover, the formed product was extracted, purified and characterised by NMR. This rigorous experimentation has lead us to the  claim that the  Isolated Neurospora sp.  is potential to produce Naringinase.  

Comment 3: Nanoparticle characterization is really poor. Authors only provide TEM images without presenting a Particle Size Distribution. PSD should be included in the paper by TEM imaging analysis or by using DLS method.

Response 3: Accepted and the new micrographs were introduced in the revised paper.

Comment 4: In a similar way the value of surface charge – Zpotential – should be measured and the stability of the particles at different times should be also included.

Response 4: Accepted and the corresponding data was shown in the supplementary material.

Comment 5: Authors should consider to study the bacteriostatic effects of NPs on liquid medium by analysing the optical density of the culture at 600 nm.

Response 5: Followed. We completely agree with the referee. However we have used the standard protocol reported that would help to find out the increase in fold activity.

Comment 6: It has been extensively reported that Ag have bactericidal effects. The increase in this property by conjugation with NAR must have another control. Not only the activity of NAR alone… The activity of silver nanoparticles conjugated with another material should be used to compare and to demonstrate that there are synergist effects because of the combination.

Response 6: We completely agree with the referee and now the data related to the activity of  Ag and Au NPs without the conjugates was added.

Comment 7: Figure 7 represents a phylogenetic tree for Neurospora based on rDNA sequence. However there is not a method for sequencing DNA on Materials section. How are these results obtained. Please clarify this misunderstanding. Please provide more details about the equipment used.

Response 7: Accepted and the revision is made.

Other minor comments:

Comment 8: Please review the citation at lines 79 to 90. Some numbers are not included in brackets.

Response 8: Accepted and corrected.

Comment 9: Line 136. Why do you express the concentration in that way?

Response 9: Accepted. No specific reason for that term. Only for simplicity the notation was followed.

Comment 10: Line 160. CentrifugeD. Spelling

Response 10:The suggested change has been made as shown in track changes mode.

Reviewer 3 Report

Comments and Suggestions for Authors

Dear Editor

Thank you for sending me the manuscript for review. This is an interesting area of study. The characterization part of the manuscript needs further improvement. Particularly, figure 5 (a-d) is not suitable for publication. A better TEM image and corresponding elemental composition graph are required for publication.

Best regards

Reviewer

Comments on the Quality of English Language

May be improved 

Author Response

We are indeed grateful to the referee for pinpointing the changes required by sharing his expertise. The paper is revised accordingly. Changes made are shown in track changes mode. A point by point response to referee's comments is provided bellow:

Report of reviewer 3:

General comments and suggestions for authors and the editor: Thank you for sending me the manuscript for review. This is an interesting area of study.

Comment 1: The characterization part of the manuscript needs further improvement.

Particularly, figure 5 (a-d) is not suitable for publication. A better TEM image and corresponding elemental composition graph are required for publication.

Response 1:We thank the referee for the useful comment. Following the suggestion, better TEM images were provided by locating the point of elemental analysis in the TEM image. Moreover the particle size distribution plots were also provided in the revised figure 5.

Comments 2: On the Quality of English Language, may be improved 

Response 2: We thank the referee for the constructive criticism and the suggestions for improvement. Following the advice, the English language has been refined throughout the paper and the changes made are shown in track changes mode.

Round 2

Reviewer 1 Report

Comments and Suggestions for Authors

The data shown in figure 6 c and d should be still revised and plotted using another software to make the graphs more readable.

Line 344 ff: Please add the standard deviation of the size measurements in the text to give the reader an idea about the monodispersity.

Figure S2a and c: Please add in the figure legend that the data was obtained by dynamic light scattering. It might be nice to add the mean values in the plot as well.

Comments on the Quality of English Language

OK

Author Response

Manuscript ID:bioengineering-2906949 R2

Reponse to referee’s comments:

Report of reviewer 1:

Comment 1:  The data shown in figure 6 c and d should be still revised and plotted using another software to make the graphs more readable.

Response 1:  Infact, the EDXA traces are machine generated plots and we do not have data. However, following the advise of the referee, figure 6 has been improved as shown on page 10, lines from 364 to 367;  now the EDAX traces were moved to the electronic supplementary information and given a separate figure name as shown on page 3 lines 43 to 48 (ESI).

Comment 2:  Line 344 ff: Please add the standard deviation of the size measurements in the text to give the reader an idea about the monodispersity.

Response 2:  The details on sample size and standard deviation values were now included in the revised paper. The changes made following the suggestion can be seen in track changes mode on page 9, lines 345 and 346.

Comment 3:  Figure S2a and c: Please add in the figure legend that the data was obtained by dynamic light scattering. It might be nice to add the mean values in the plot as well.

Response 3:  We thank the referee for evaluating the ESI Following the suggestion, figure S2 a and c have been revised.  The changes made can be see on page 2, line 42 – 44 (ESI) in track changes mode.

Reviewer 2 Report

Comments and Suggestions for Authors

Authors have followed my recommendations and now I recommend the paper for publication.

Author Response

Report of reviewer 2:  Authors have followed my recommendations and now I recommend the paper for publication.

Response to reviewer 2: We are deeply indebted to the reviewer for the valuable insight that has improved the quality of the paper. Moreover, the report was provided much earlier than expected.

Reviewer 3 Report

Comments and Suggestions for Authors

Dear Editor

Thank you for sending me the manuscript. The manuscript is interesting. However, characterization part  needs improvement. Elemental composition analysis in Figure 6(c) is not suitable for publication. Author may perform another elemental composition study independently.

Further, antimicrobial study is not sufficient. Detailed study and pictures are required.

Manuscript may be considered for publication after major revision.

Best regards

Reviewer

Comments on the Quality of English Language

Moderate correction

Author Response

Report of  Reviewer 3:

Manuscript may be considered for publication after major revision.

Comment 1: Thank you for sending me the manuscript. The manuscript is interesting. However, characterization part  needs improvement. Elemental composition analysis in Figure 6(c) is not suitable for publication. Author may perform another elemental composition study independently.

Response 1:   We are indeed thankful to the referee for the critical insights for improving the standard of our paper.  The elemental composition analysis is not deleted from the main paper and shown in the electronic supplementary material.  The changes made in figure 6, following the suggestion of the referee can now be seen in track changes mode on page 10, lines from 364 to 367;  

Comment 2:  Further, antimicrobial study is not sufficient. Detailed study and pictures are required.

Response 2:   Additional antimicrobial studies were performed and the new data and results added were shown in track changes mode. The changes made can now be seen on page 11   lines 383 – 387.

Comment 3  on the Quality of English Language: Moderate correction

Response 3:  Following the suggestion of the referee, the paper has been edited for English language and the changes made were shown in track changes mode throughout the paper.

Round 3

Reviewer 3 Report

Comments and Suggestions for Authors

Dear Editor

Thank you again for sending me the manuscript for review. The author has improved the manuscript. However, still characterization part still needs improvement. I did not see any change in TEM images. Please improve the TEM image and add EDS data instead of IR. IR may be sent to supplementary. Please add MIC and MBC in the antibacterial part.

A major revision is required for publication.

Best regards

Reviewer

Comments on the Quality of English Language

Can be improved

Author Response

Manuscript ID:bioengineering-2906949 R2

Report of Reviewer 3:

General comment: A major revision is required for publication.

Comment 1: The author has improved the manuscript. However, still characterization part still needs improvement.

Response 1: We are thankful the referee for the useful suggestions that facilitated the improvement.   The characterization part too has been improved in the revised paper following the suggestions and the changes made were shown in track changes mode.

Comment 2: I did not see any change in TEM images. Please improve the TEM image and add EDS data instead of IR. IR may be sent to supplementary.

Response 2: We thank the referee for the useful advice and such a change has been made as shown in Figure 6 on page 11 in the manuscript and on page 5 in the electronic supplementary information (ESI).

Comment 3: Please add MIC and MBC in the antibacterial part.

Response 3:  We thank the referee for the advice and the result on such studies has been added as shown in lines 383-385 in the manuscript and on pages 5 and 6 in the ESI.

Comment 4: on the Quality of English Language: Can be improved.

Response 4: Following the advice of the referee, the English language has been improved and the corresponding changes made have been marked in track changes mode throughout the paper.

Round 4

Reviewer 3 Report

Comments and Suggestions for Authors

Dear Editor

Thank you again for sending me the manuscript. The authors have changed the TEM figure. MIC and MBC have been measured, however, not discussed in the text, only mentioned in the text. An appropriate discussion of the result is required. There are many typos/English errors, that need correction both in the text and supplementary. 

Best regards

Reviewer

Comments on the Quality of English Language

Need improvement
